# Electrosynthesis of formamide from methanol and ammonia under ambient conditions

Nannan Meng [1,6], Jiang Shao[1,6], Hongjiao Li [2,6], Yuting Wang [1,3], Xiaoli Fu [4], Cuibo Liu [1], Yifu Yu [1,3] ✉ & Bin Zhang [1,3,5] ✉

Electrochemical conversion of abundant carbon- and nitrogen-containing small molecules into high-valued organonitrogen compounds is alluring to reducing current dependence on fossil energy. Here we report a single-cell electrochemical oxidation approach to transform methanol and ammonia into formamide under ambient conditions over Pt electrocatalyst that provides 74.26% selectivity from methanol to formamide and a Faradaic efficiency of 40.39% at 100 mA cm$^{-2}$ current density, gaining an economic advantage over conventional manufacturing based on techno-economic analysis. A 46-h continuous test performed in the flow cell shows no performance decay. The combined results of in situ experiments and theoretical simulations unveil the C–N bond formation mechanism via nucleophilic attack of $NH_3$ on an aldehyde-like intermediate derived from methanol electrooxidation. This work offers a way to synthesize formamide via C–N coupling and can be extended to substantially synthesize other value-added organonitrogen chemicals (e.g., acetamide, propenamide, formyl methylamine).

Amides, a very important class of compounds in chemistry and biology, have been studied extensively over the past century[1–3]. Formamide alone has an annual global market of millions of tons[4]. Currently, formamide is produced through the reaction of carbon monoxide and ammonia under high-temperature and high-pressure conditions via the following two strategies (Eqs. (1) and (2), Fig. 1a)[5], which consumes huge fossil fuels and aggravates the greenhouse effect. Searching for novel solutions that allow energy-efficient and green synthesis of formamide is significant. The electrochemical technique, especially driven by renewable energy, has gained increasing attention for the synthesis of many high-valued chemicals[6–14]. For instance, the electrosynthesis of methylamine, formamide, and acetamide has been successfully achieved by using $CO_2$/CO as the carbon source[6–8,10]. At present, these important advances mainly focused on the electrochemical reduction reactions to construct the C–N bond. But, the sluggish anodic oxygen evolution reaction requires a high applied potential. Thus, the exploration of an

alternative electrooxidation process using abundant carbon- and nitrogen-containing feedstocks to synthesize formamide under ambient conditions is attractive but remains a great challenge.

$$CO + NH_3 \xrightarrow[p=0.8-1.7\text{MPa},\, T=348-353\text{K}]{CH_3ONa} HCONH_2 \qquad (1)$$

$$CO + CH_3OH \xrightarrow[p=10-30\text{MPa},\, T=353-373\text{K}]{CH_3ONa} HCOOCH_3 \qquad (2)$$

$$HCOOCH_3 \xrightarrow[p=0.1-0.3\text{MPa}\, T=313-373\text{K}]{} HCONH_2$$

Methanol with the reputation of "liquid sunshine" can be mass-produced via carbon dioxide ($CO_2$) reduction and biomass conversion[15–19]. The development of green chemical reactions to upgrade $CO_2$-/biomass-derived methanol into high-valued chemicals

[1]Department of Chemistry, Institute of Molecular Plus, Tianjin University, Tianjin 300072, China. [2]School of Chemical Engineering, Sichuan University, Chengdu, Sichuan 610065, China. [3]Haihe Laboratory of Sustainable Chemical Transformations, Tianjin 300072, China. [4]School of Earth System Science, Tianjin University, Tianjin 300072, China. [5]Tianjin Key Laboratory of Molecular Optoelectronic Sciences, Key Laboratory of Systems Bioengineering (Ministry of Education), Tianjin University, Tianjin 300072, China. [6]These authors contributed equally: Nannan Meng, Jiang Shao, Hongjiao Li. ✉e-mail: yyu@tju.edu.cn; bzhang@tju.edu.cn

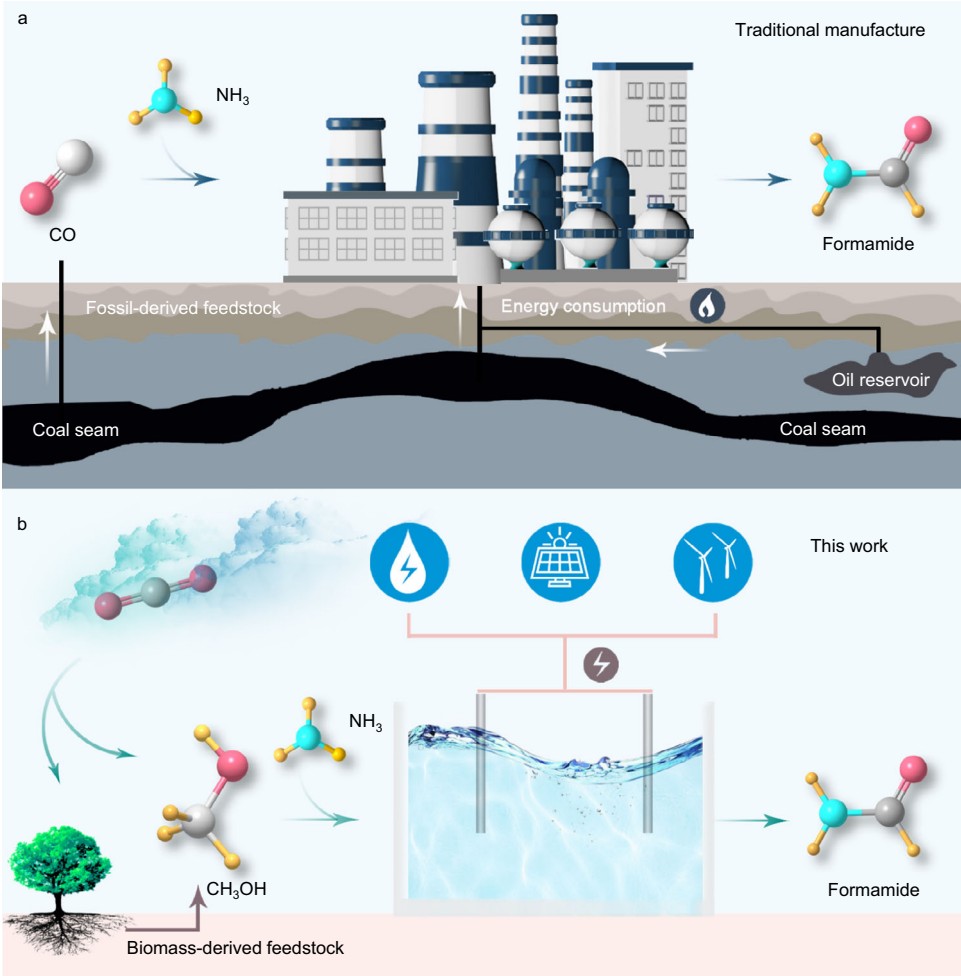

**Fig. 1 |** The manufacturing approaches of formamide by the industrial thermochemical system (**a**) and the proposed electrochemical system (**b**).

can contribute to greenhouse gas mitigation for a carbon-neutral future. Notably, direct thermal-catalytic oxidation of aromatic alcohol using over-stoichiometric oxidants to form an aldehyde-like intermediate and react with ammonia under high pressure was proved to prepare aromatic amides by Noritaka Mizuno and co-workers[20]. Therefore, we propose that the utilization of nucleophilic attack of $NH_3$ on the in situ formed formaldehyde-like intermediate from methanol electrooxidation may afford a greener process to produce formamide under ambient conditions.

Herein, we demonstrate a methanol electrolysis approach to synthesize formamide in an aqueous ammonia medium at ambient temperature and pressure (Fig. 1b). Among all the screened electrocatalysts, Pt shows the highest performance. The optimized selectivity from methanol to formamide and Faradaic efficiency can reach 74.26% and 40.39% at 100 mA cm$^{-2}$ current density in a single cell. The key reaction intermediates are recognized by isotope-labeled in situ Attenuated Total Reflection Flourier Transformed Infrared Spectroscopy (ATR-FTIR) and online differential electrochemical mass spectrometry (DEMS). Combining the computational study, the high formamide production efficiency is ascribed to the moderate binding affinity of the reaction intermediates on $PtO_2$, which is formed on the surface of the Pt electrocatalyst during the reaction. Furthermore, a flow cell is adopted for continuous formamide electrosynthesis without performance decay in a 46-h stability test. Notably, the techno-economic analysis (TEA) proves the cost advantage of formamide electrosynthesis strategy over current industry manufacturing.

## Results

The conjecture of formamide electrosynthesis from methanol and ammonia oxidation is testified in a membrane-free single cell. The cathode is metal Ni. Ten species of metal anode catalysts, including Pt, Ni, Fe, Cu, Al, Co, Ti, Pb, Mo, and W, were screened using the galvanostatic method. A single cell can greatly reduce the operating cost compared with a membrane-separated two-chamber cell. The mixture of methanol and ammonia with a 2:1 volume ratio in 0.5 M $NaHCO_3$ aqueous solution was measured at the current density of 10 mA cm$^{-2}$. The carbonaceous liquid product was analyzed and quantified by $^1$H-nuclear magnetic resonance ($^1$H-NMR, Supplementary Fig. 1). After 3-h electrolysis, Pt, Ni, and Fe catalysts show the capacity for formamide formation, and their Faradaic efficiencies (FE$_{formamide}$) are 11.70%, 7.31%, and 1.44% (Supplementary Fig. 2a), respectively. Among them, Pt delivers the main carbonaceous liquid product of formamide while Ni and Fe mainly produce formic acid products (Supplementary Fig. 2b). At different current densities (20, 40, 80, 100, 120, and 150 mA cm$^{-2}$), Pt exhibits higher Faradaic efficiency and yield rate for formamide compared with Ni and Fe (Fig. 1a and Supplementary Figs. 3, 4). The optimized FE$_{Formamide}$ over Pt reaches 32.70% at the current density of 100 mA cm$^{-2}$. This FE$_{Formamide}$ value corresponds to the yield rate of 305.4 μmol cm$^{-2}$ h$^{-1}$, greatly higher than Ni (4.93%, 46.03 μmol cm$^{-2}$ h$^{-1}$) and Fe (1.43%, 13.36 μmol cm$^{-2}$ h$^{-1}$) (Supplementary Fig. 5). To cut the cost of catalyst, Pt-covered Ti foil was synthesized by electrodeposition of Pt on Ti foil for replacing bulk Pt foil (Supplementary Fig. 6). Pt-Ti shows a similar performance to bulk

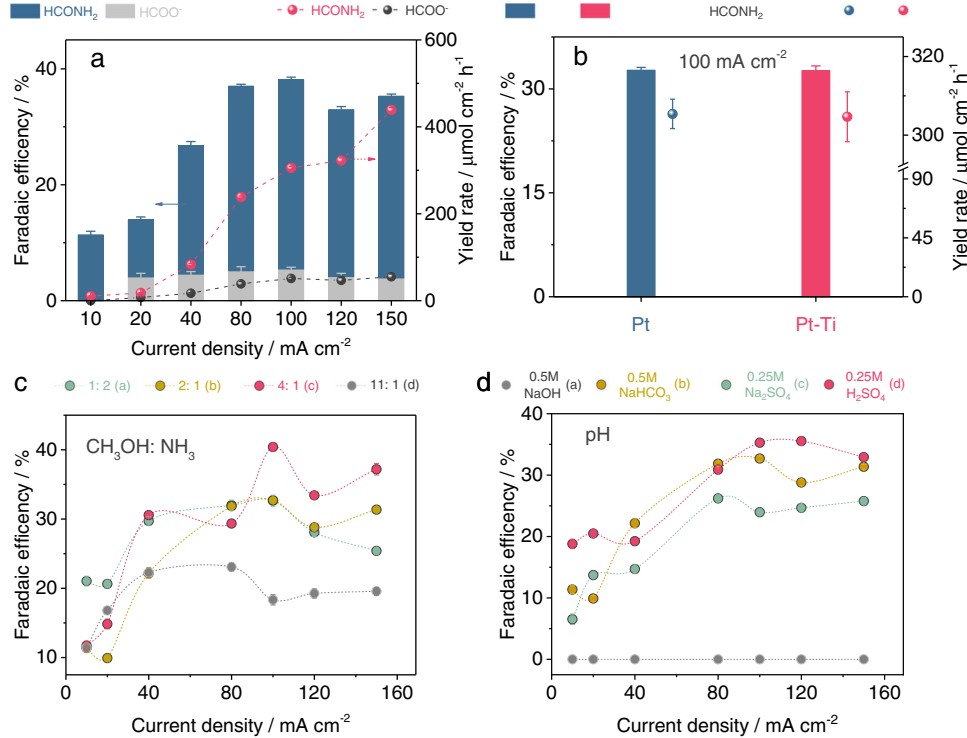

**Fig. 2 | Formamide electrosynthesis from methanol and ammonia oxidation.** **a** The optimization of the current densities over Pt for formamide electrosynthesis. **b** Formamide electrosynthesis performance over Pt and Pt-Ti at 100 mA cm⁻². The current density-dependent electrosynthesis performance over Pt-Ti with different **c** volume ratios of $CH_3OH$ to $NH_3$ and **d** pH values of electrolytes. Error bars correspond to the Standard Deviation (SD) of three independent measurements.

Pt for formamide generation at 100 mA cm⁻² (Fig. 2b and Supplementary Fig. 7).

## The performance optimization and stability measurement of Pt-Ti catalyst

The $FE_{formamide}$ increases firstly and then decreases with increasing the volume ratio of $CH_3OH$ to $NH_3$ from 1:1 to 11:1. The optimized $FE_{formamide}$ reaches 40.39% at 4:1 at 100 mA cm⁻² and the corresponding selectivity from methanol to formamide was 74.26% (Fig. 2c and Supplementary Fig. 8). The carbonaceous liquid product is only formic acid and formamide is completely suppressed in 0.5 M NaOH aqueous solution (Fig. 2d and Supplementary Fig. 9). After reducing the alkalinity, formamide emerges. The Faradaic efficiency of formamide reaches the maximum in the electrolyte of 0.5 M NaHCO₃ or 0.25 M H₂SO₄. The stability of Pt-Ti catalyst for formamide electrosynthesis is examined at 100 mA cm⁻² in 0.5 M NaHCO₃. The Faradaic efficiency and yield rate keep stable for over 10 cycle tests (Supplementary Fig. 10). 1.1 ng s⁻¹ dissolution of Pt is quantified by using the inductively coupled plasma emission spectrometer (ICP). Although a small amount of Pt dissolution is induced by oxidation, SEM and XRD of used Pt-Ti samples still show no obvious change. And slight surface oxidation of used Pt is confirmed by the XPS spectrum of $Pt^{4+}$ $4f_{5/2}$ signal peak at 78.1 eV (Supplementary Fig. 11)[21]. In addition, this approach can be utilized in the synthesis of acetamide, propenamide, and formyl methylamine, suggesting the expandability of our methodology (Supplementary Fig. 12).

## Mechanistic studies

To explore the reaction pathway, Density Functional Theory (DFT) is carried out. For building proper theory models, the surface component of the catalyst should be confirmed during the reaction. Electrochemical in situ Raman spectroscopy (Supplementary Fig. 13), a surface-sensitive technique, is adopted to trace the active phases of

those catalysts during the reaction process. As shown in Fig. 3a–c, with increasing the current density, the peak intensities of $\alpha$-$PtO_2$, $\beta$-NiOOH, and $\alpha$-FeOOH on Pt, Ni, and Fe can be identified[22–24]. These results indicate that the active phases of Pt, Ni, and Fe for formamide electrosynthesis are $\alpha$-$PtO_2$, $\beta$-NiOOH, and $\alpha$-FeOOH, respectively. Therefore, $\alpha$-$PtO_2$, $\beta$-NiOOH, and $\alpha$-FeOOH are used to build the surface models for DFT simulations. The complete reaction pathway and energy diagram of the coupling reaction as well as the electronic analysis of the C−N bond formation steps are displayed in Fig. 3d, Supplementary Figs. 14–20 and Supplementary Tables S1–S5. There are three pathways for amide formation using alcohol and ammonia as the feedstocks on $\alpha$-$PtO_2$: (Path 1) aldehyde from alcohol dehydration reacted with $NH_3$ to form hemiaminal that is subsequently dehydrated to formamide[25–28]; (Path 2) aldehyde as the intermediate of methanol oxidation reacts with $NH_3$ to form aldimine via a hemiaminal intermediate, and then the aldimine is oxidized to nitrile that can be further hydrolyzed to formamide[20]; (Path 3) alcohol-derived CHO* reacts with $NH_3$-derived $NH_2$* to generate formamide (Supplementary Fig. 14, Supplementary Table 1). For Path 3, the C−N coupling requires two adjacent active intermediates of CHO* and $NH_2$*, which are both easy to be solely oxidized into the corresponding carbon-/nitrogen-containing byproducts[29,30]. Thus, the direct nucleophilic attack of $NH_3$ on the in situ formed aldehyde in Paths 1 and 2 seems more possible (Fig. 3d). As for formamide formation on $\beta$-NiOOH and $\alpha$-FeOOH, only Path 1 is possible (Fig. 3d, Supplementary Figs. 15–18, Supplementary Tables 2, 3). The key C−N coupling step is then analyzed from a kinetic point (Supplementary Fig. 19, Supplementary Table 4). The C−N bond formation between *$CH_2O$ and $NH_3$ is the nucleophilic attack process, i.e., the positively charged C is attacked by the electronegative N atom in $NH_3$[20,25]. Thus, the charged states of C in *$CH_xO$ and N in $NH_3/NH_2$ qualitatively demonstrate the feasibility of the C−N bond-making process. The charge analysis of the relevant adsorbates is done using Bader charge analysis. As shown in Supplementary Table 4, C in *$CH_2O$ is positively charged in the order of +1.60 e on $\alpha$-$PtO_2$ > +1.36 e on

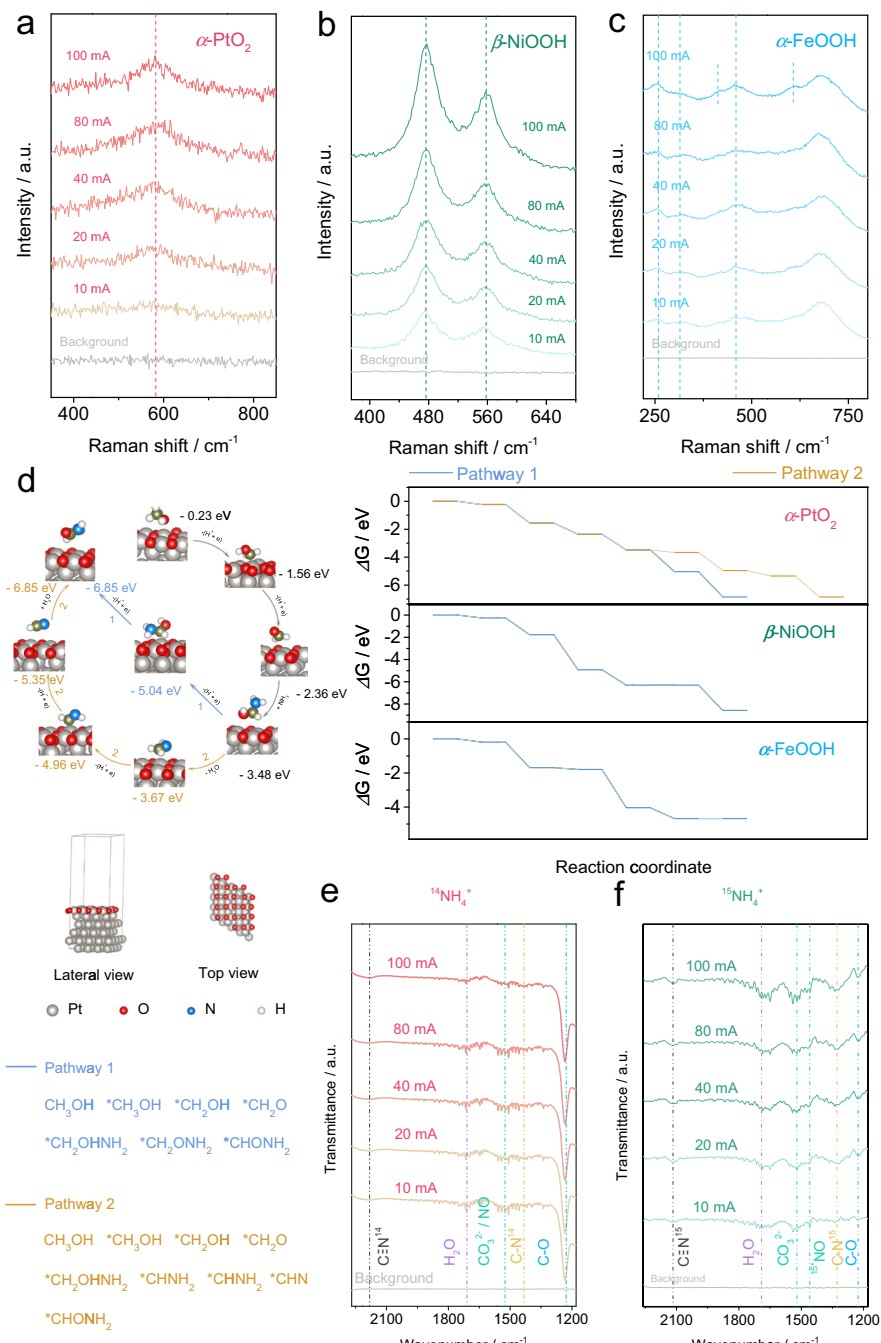

**Fig. 3 | The superficial active sites, mechanism, and reaction intermediates analysis. a–c** The current density-depended Raman signals of Pt, Ni, and Fe. **d** The theoretical model of $\alpha$-PtO$_2$, formamide formation pathway over its surface and free energy diagram of HCONH$_2$ formation over $\alpha$-PtO$_2$, $\beta$-NiOOH, and $\alpha$-FeOOH via different pathways. **e, f** Isotope-labeled in situ ATR-FTIR measurements using $^{14}$NH$_4^+$ and $^{15}$NH$_4^+$.

$\beta$-NiOOH > +0.20 e on $\alpha$-FeOOH. We, therefore, propose that the barrier energies for the C−N bond coupling process by NH$_3$ nucleophilic attack of *CH$_2$O are very likely to be in the order of $\alpha$-PtO$_2$ > $\beta$-NiOOH > $\alpha$-FeOOH. For Path 1, the minimum applied potentials demanded to make all elementary reactions exoergic are 1.00, 2.51, and 2.57 V for $\alpha$-PtO$_2$, $\beta$-NiOOH, and $\alpha$-FeOOH, respectively (Supplementary Table 5). $\alpha$-PtO$_2$ performs an obviously stronger catalytic activity toward the coupling of methanol and ammonia, which is rooted in its higher ability of oxidizing methanol and stabilizing *CH$_2$ONH$_2$. It should be noted that $\alpha$-PtO$_2$ also benefits the suppression of the complete methanol oxidation due to its smallest driving force of the *CH$_2$O-*CHO step (Fig. 3d, Supplementary Table 5). For Path 2, the

crucial reaction step of *CH$_2$OHNH$_2$ dehydration only thermodynamically takes place on $\alpha$-PtO$_2$ with Gibbs free energy of -0.19 eV, which excludes the possibility of Path 2 on $\beta$-NiOOH and $\alpha$-FeOOH (Supplementary Table 5). Thus, $\alpha$-PtO$_2$ facilitates the formamide formation via Path 1 and provides one more possible Path 2.

To confirm the reaction pathway, we first testify the methanol electrolysis without ammonia. The carbonaceous liquid product is only formic acid. Then, no other carbonaceous liquid product is detected when the mixture of formic acid and ammonia is electrolyzed. In addition, an H-type electrolysis experiment proves the cathode reaction does not affect formamide formation (Supplementary Fig. 20). Those results indicate that formamide is formed from the

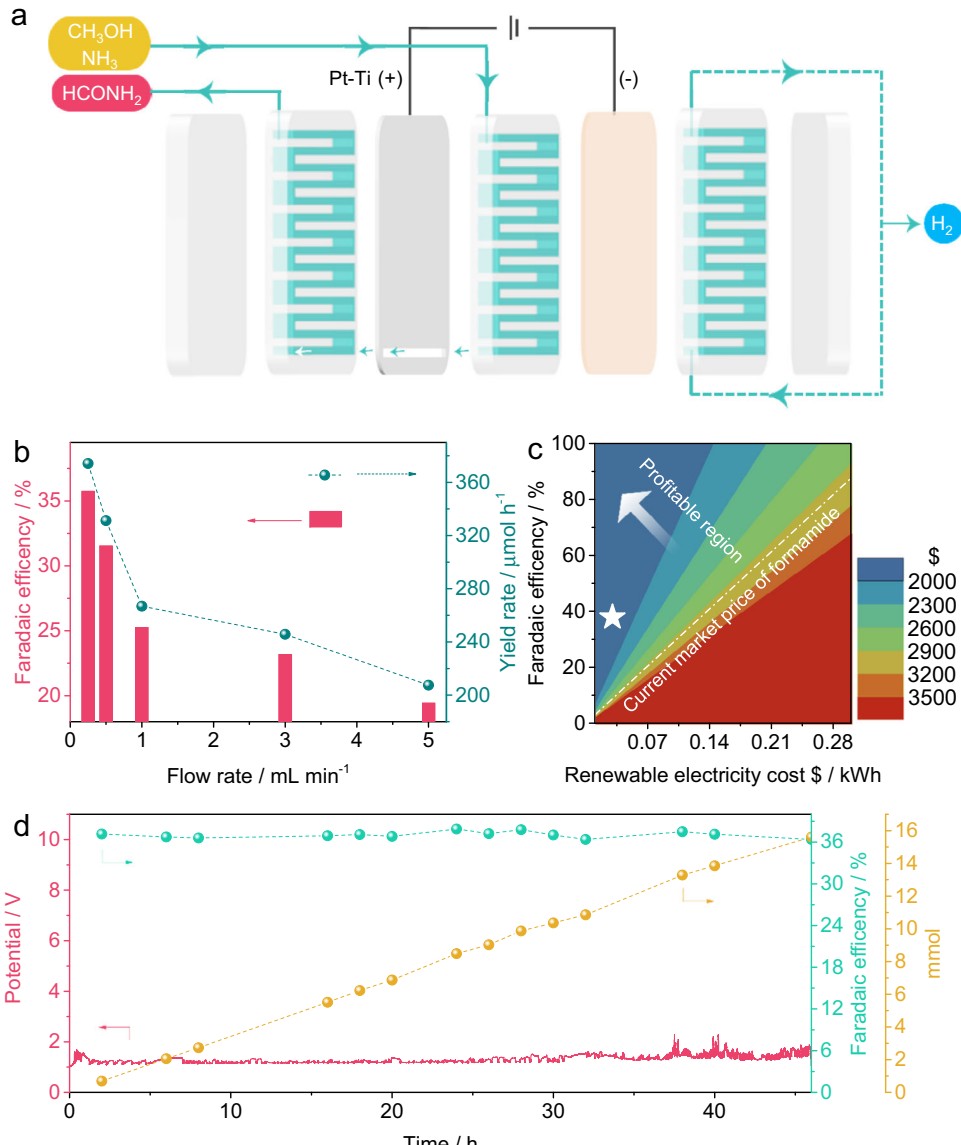

**Fig. 4 | The continuous electrosynthesis of formamide in a flow cell. a** The scheme of flow cell; **b** The optimization of flow rate for formamide electrosynthesis; **c** TEA of the levelized cost of formamide electrosynthesis as a function of Faradaic efficiency and renewable energy cost; **d** Cell potential, Faradaic efficiency, and the output of formamide over 46 h.

combination of the intermediate from methanol-to-formic acid conversion (formaldehyde-like intermediate) and ammonia. Formaldehyde intermediate can be quickly converted to formic acid during the electrolysis without ammonia, which makes it undetectable. When methanol feedstock is replaced with formaldehyde, formamide can be detected, further proving the existence of an aldehyde-like intermediate (Supplementary Table 6). These results are consistent with theoretical simulation. To further understand and real-time monitor the catalytic process, the cutting-edge in situ ATR-FTIR is carried out in a transmission mode, suggesting the product signal in the form of the downward peak during the measurement process (Supplementary Fig. 21). Using $^{14}NH_4^+$ as the nitrogen source, the signal shows an increasing tendency with the current density (Fig. 3e). The characteristic peaks at around 1220, 1435, 1520, 1680, and 2185 cm$^{-1}$, corresponding to C–O, C–N, CO$_3^{2-}$/NO, H$_2$O, and C≡N, can be identified[31–35], suggesting the production of multiple intermediates. For confirming the C–N bond formation, isotope-labeling in situ ATR-FTIR is carried out using $^{15}NH_4^+$ (Fig. 3f). By applying the increasing current density, C–$^{14}$N and C≡$^{14}$N shift to the lower wavenumber region

(1360 cm$^{-1}$ for C-$^{15}$N, and 2120 cm$^{-1}$ for C≡$^{15}$N) because of the isotope effect[33,36]. NO signal at around 1460 cm$^{-1}$ is also observed[37]. Moreover, a series of control experiments prove the Au substrate and single CH$_3$OH/NH$_3$ electrooxidation do not disturb the detection of C–N and C≡N signals in FTIR spectra (Supplementary Figs. 22, 23). C=N in H$_2$C=NH is traced based on online DEMS (Supplementary Fig. 24)[38]. Accompanied by the on/off switching circuit (Supplementary Fig. 25a), the signal of H$_2$C=NH raises and falls, implying its generation. The 46 molecular weight of formamide product (molecular weight should plus 1 in the positive ion mode) is identified by the liquid chromatography-mass spectrometry (LC-MS) (Supplementary Figs. 25b and 26). The combined results of isotope-labeling in situ ATR-FTIR and online DEMS confirm the existence of Path 2 for formamide generation on α-PtO$_2$.

For realizing its continuous production, a flow cell is designed for the electrosynthesis of formamide. CH$_3$OH and NH$_3$ as the feedstocks flow along one side of the Pt-Ti anode, then go through its slit, and are finally attached to the other side of Pt-Ti with H$_2$ evolution from the cathode reaction (Fig. 4a). According to the above-mentioned

optimized condition, 0.5 M NaHCO$_3$ and 4:1 volume ratio of CH$_3$OH to NH$_3$ are adopted. Different flow rates from 0.25 mL min$^{-1}$ to 5 mL min$^{-1}$ are carried out. As shown in Fig. 4b, FE for formamide increases with decreasing the flow rate and the maximum value can reach 37.88% at 0.25 mL min$^{-1}$, corresponding to 353.35 μmol h$^{-1}$ of yield rate. A techno-economic analysis (TEA, Supplementary Note 1, Supplementary Fig. 27) indicates the profit per tonne of formamide from this electrosynthesis can reach $1325.67 (marked as a star in Fig. 4c) and as high as $2158.90 with adding the other dividends such as the benefits of hydrogen, sodium formate, sodium nitrate, and sodium nitrite. After 46 h of continuous operation, the catalyst can maintain its performance (Fig. 4d).

## Discussion

A sustainable electrooxidation approach is reported to synthesize formamide using methanol and ammonia as the feedstocks at ambient temperature and pressure. After screening 10 metal catalysts, Pt exhibits the highest activity for formamide synthesis. The maximum Faradaic efficiency reaches 40.39% at the current density of 100 mA cm$^{-2}$ in NaHCO$_3$ solution. The combined results of theoretical simulation and isotope-labeled in situ electrochemical characterizations unveil the reaction pathway for formamide formation and the high activity origin of Pt catalyst. The formamide product is further confirmed by isotope-labeled LC-MS. This work presents a novel synthesis approach to formamide using the CO$_2$-derived feedstock under mild conditions as a promising alternative to thermal chemical manufacturing with fossil-supported feedstock and energy. Impressively, this facile method is also suitable for the synthesis of acetamide, propenamide, and formyl methylamine, highlighting the promising application potential.

## Methods

### Materials

All reagents and metal catalysts (Pt, Ni, Fe, Cu, Al, Co, Ti, Pb, Mo, W foils, and Pt powder) were obtained from commerce without further purification. CH$_3$OH, HCHO, HCN, dimethyl sulphoxide (DMSO), HCOONa, HCONH$_2$, NaHCO$_3$, Na$_2$SO$_4$, (NH$_4$)$_2$SO$_4$, NaOH, NH$_4$F, NaAuCl$_4$·2H$_2$O, K$_2$PtCl$_4$, NH$_4$Cl, Na$_2$SO$_3$, and Na$_2$S$_2$O$_3$·5H$_2$O are analytical grade. The concentrations of NH$_3$, H$_2$SO$_4$, HNO$_3$, HCl, H$_2$O$_2$, D$_2$O, and HF are 28%, 98%, 68%, 38%, 30%, 99% and 40%, respectively. The isotope abundance isotope-labeling ($^{15}$NH$_4$)$_2$SO$_4$ is 98.5%.

### Synthesis of Pt-coated Ti substrate (Pt-Ti)

Pt-Ti was synthesized by the electro-deposition method. First, 1 × 3 cm$^2$ Ti foil was etched in H$_2$SO$_4$ aqueous solution ($V_{concentrated\ H2SO4}:V_{H2O} = 1:2$) at 70 °C for 20 min. After washing three times with water, Ti substrate was obtained. Second, Ti substrate, carbon rod, and saturated calomel electrode (SCE) were placed in 25 mM K$_2$PtCl$_4$/200 mM HCl aqueous solution and served as the cathode, anode, and reference electrode, respectively. Finally, the applied voltage was set as -0.1 V (vs. SCE) for 5 min. After washing three times with water, Pt-Ti substrate was obtained. The mass loading of Pt on Ti surface is controlled at 2.0 ± 0.1 mg cm$^{-2}$.

### Material characterization

Scanning electron micrograph (SEM) image and the corresponding energy dispersive X-ray (EDX) spectrum were performed on a Regulus 8100 field emission scanning electron microscopy. X-ray diffractometry (XRD) pattern was carried out on a Bruker D8-Focus instrument. X-ray photoelectron spectroscopy (XPS) spectrum was collected on a Thermo Fisher Scientific K-Alpha+ instrument.

### Electrochemical measurements

CH$_3$OH and ammonia with different volume ratios were added into a 0.5 M NaHCO$_3$ aqueous solution. Ni foil, Pt-Ti, and Ag/AgCl were inserted in the solution and as the cathode, the anode, and the reference electrode, respectively. Finally, different current densities from 10 to 150 mA cm$^{-2}$ were carried out using the electrochemical workstation (CS150H, Wuhan CorrTest Instruments Co., Ltd) and the reaction time was set as 3 h. The influence of the reactant ratio on the catalytic performance was performed by controlling the total volume of 15 mL (the mixture of CH$_3$OH and ammonia) and other experimental conditions were kept consistent. The influence of anion on the catalytic performance was performed by changing the different electrolytes and controlling the total molar of the mixture of the anion and the cation in different electrolytes was the same.

### Product quantification

The gaseous products were quantified using gas chromatography (Agilent 7890A) equipped with thermal conductivity detection (TCD) and flame ionization detection (FID). High-purtily He was employed as the carrier gas. The carbonaceous liquid products were analyzed by $^1$H nuclear magnetic resonance ($^1$H-NMR) using the DMSO as the internal standard. The preparation of the internal standard for $^1$H-NMR detection was as follows: 10 μL DMSO was diluted 100 times by water and then mixed with D$_2$O with a 1:1 (V/V) ratio. The calibration curves of the carbonaceous liquid products were obtained by plotting the standard sample concentration vs. the $^1$H-NMR peak area ratio of the standard sample/DMSO (Supplementary Fig. S1). Formamide was further identified by a liquid chromatography-tandem mass spectrometry (LC-MS) (SCIEX 6500 PLUS) using a Phenomenex NH$_2$ column. The parameters for LC-MS detection were set as follows: the aqueous solution with acetonitrile/water (40:60, V/V) was used as the mobile phase. The injected quantity, flow rate, and detection wavelength were 10 μL, 0.2 mL min$^{-1}$, and 195 nm, respectively. The data collection was recorded in the positive ion mode. Nitrate and nitrite were analyzed by ion chromatography (IC). The parameter for IC detection was set as follows: 25 mM KOH was used as the mobile phase. The injected quantity, flow rate, and column temperature were 25 μL, 1 mL min$^{-1}$, and 30 °C, respectively.

### Online differential electrochemical mass spectrometry (DEMS) measurement

Online DEMS was carried out to detect the volatile matters during the real-time electrolysis process (Shanghai Linglu Instrument & Equipment Co). The signal was collected through a hydrophobic polytetrafluoroethylene (PTFE) membrane, which played a key role in permitting the volatile matter and simultaneously preventing water into the vacuum chamber. The electrolysis reaction occurred on one side of the PTFE membrane and the produced volatile matters were brought to the other side through a pump (Supplementary Fig. S24). Sequentially, those matters were detected by mass spectrometry.

### In situ attenuated total reflection flourier transformed infrared spectroscopy (ATR-FTIR) measurement

In situ ATR-FTIR was carried out to trace the signals of the intermediates using a Nicolet Nexus 670 Spectroscopy equipped with a liquid nitrogen-cooled mercury-cadmium-telluride (MCT) detector. An ECIR-II cell equipped with a Pike Veemax III ATR in a three-electrode system was provided from Shanghai Linglu Instrument& Equipment Co. The data was collected through the Pt-covered monocrystal silicon. To improve the signal intensity, the monocrystal silicon was initially coated with a layer of Au using the chemical plating method as follows: (1) 0.12 g NaOH, 0.23 g NaAuCl$_4$·2H$_2$O, 0.13 g NH$_4$Cl, 0.95 g Na$_2$SO$_3$, 0.62g Na$_2$S$_2$O$_3$·5H$_2$O were dissolved in 100 mL H$_2$O (denoted as Solution A); (2) The monocrystal silicon was immersed in the aqua regia ($V_{concentrated\ HCl}:V_{HNO3} = 1:1$) for 20 min and then polished using the Al powder for 10 min. After washing three times with water, the clean monocrystal silicon was obtained; (3) The above monocrystal silicon was immersed in the mixture of H$_2$SO$_4$ and H$_2$O$_2$ ($V_{concentrated\ H2SO4}:V_{H2O2} = 1:1$) for

20 min; (4) After washing three times with water, the above mono-crystal silicon was then immersed in 40% $NH_4F$ aqueous solution and washed three times with water; (5) The monocrystal silicon was then immersed in the mixture of 15 mL solution A and 3.4 mL 2% $NH_4F$ aqueous solution; (6) After 5 min, Au coated monocrystal silicon was obtained. To identify the production of the C−N bond, the isotope-labeled $(^{15}NH_4)_2SO_4$ was used as the electrolyte. To keep conditions consistent, $(^{14}NH_4)_2SO_4$ was also performed.

**In situ electrochemical Raman spectroscopy measurement**
In situ Raman was carried out to trace the surface transformation of the electrocatalysts using a Renishaw inVia reflex Raman microscope equipped with an excitation of 633 nm laser. The configuration was described in our recent works[39–42]. Different current densities were controlled by an electrochemical workstation.

## Data availability
The data that support the plots within this paper are available from the corresponding author upon reasonable request. The source data underlying Figs. 2–4 are provided as a Source Data file. Source data are provided with this paper.

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

## Acknowledgements

The authors are grateful to the National Natural Science Foundation of China (Nos. 22271213 (B.Z.), 22109115 (N.M.), and 22071173 (Y.Y.)).

## Author contributions

B.Z. conceived the idea and directed the project. B.Z., Y.Y., and N.M. designed the experiments. N.M. and J.S. carried out the materials synthesis and characterization. H.L. contributed to the theoretical calculation. Y.W. assisted in electrochemical in situ ATR-FTIR. X.F. contributed to LC-MS. C.L. discussed the mechanism. N.M. and H.L. co-wrote the paper. Y.Y. and B.Z. revised the manuscript. All authors discussed the results and commented on the manuscript.

## Competing interests

The authors declare no competing interests.
