## [Peer review file · Nature Communications]

REVIEWER COMMENTS

Reviewer #1 (Remarks to the Author):

In the submitted work, the group of Bin Zhang demonstrated the oxidative synthesis of formamide from NH₃ and methanol building blocks. Formamide electrosynthesis was accomplished with Pt/PtO₂ catalysts (after screening other candidates) with a high Faradaic efficiency of 40%. Mechanistic studies proposed that the key step is a nucleophilic attack by the NH₃ onto a formaldehyde like surface intermediates (originating from methanol). Finally, a techno-economic analysis was performed to point to the profitability of this route.

The area of heterogeneous electrosynthesis beyond water/CO₂ electrolysis is rapidly growing and the discovery of new reaction pathways such as this can be a very important boost to this end. I appreciate the integration of techno-economic analysis with the theoretical and experimental efforts to drive home the importance of this. That being said, there are several issues/points to be addressed prior to consideration for publication.

Infrared spectroscopy: The experiments here are only partially convincing. I do not understand why only 10% enriched ¹⁵NH₃ was used as 98/99% NH₃ is a readily available and cheap reagent. This should be used in place for definitive peak assignments in figure 3. Moreover, the signal quality in figure 3a especially is not sufficient for peak assignment.

If nanostructured Au was used as the substrate and I assume Pt was somehow deposited overtop, how is the activity attributed to only the Pt when Au may be exposed to the electrolyte as well. In other words, adsorbed intermediates and changes in solution that give rise to IR bands can stem from the underlying substrate as well.

Additional controls should be performed with IR measurements such as spectra taken in the absence of methanol or NH₃.

As a reference experiment, a two chamber (e.g. H-Cell) setup should be tested as well to exclude any effects from the cathode reactivity.

Regarding the stability, the electrolyte should be tested for dissolved Pt after prolonged electrolysis and the electrode should be characterized with SEM/XRD/TEM after as well.

A schematic of the Raman experiment/reaction cell should be included in the SI.

Regarding the measurements at low pH – formamide tends to decompose to formate and NH₃ in these conditions. Could the authors be underestimating their formamide production? The same goes for high pH, though there isn't any significant formamide production there to begin with. Is there a mechanistic explanation for the lack of formamide generation at high pH?

Does formamide form when starting from formaldehyde as a reagent? This could be important for mechanistic explanations. The same goes with formate.

The authors cited a work detailing acetamide formation from CO and NH₃ electrolysis. I would also recommend referencing works that discussed formamide production such as: *Chemical Science*, 13, 3957-3964 (2022)

In general, a discussion regarding previous C-N bond formation in electrochemistry would be useful to add in the introduction as well as some text regarding how this work conceptually advances the state of the art of our understanding of this process.

Regarding the scope of this scheme, if another amine was used in place of ammonia, could the same nucleophilic attack and C-N bond formation occur?

For the mechanism proposed by the authors, is cyanide ever detected in the solution as it is an intermediate. What about methylamine? The full NMR spectra should be shown, not only the formate/formamide region.

Reviewer #2 (Remarks to the Author):

In this work, Zhang et al. developed a sustainable electrooxidation approach to synthesize formamide using methanol and ammonia as feedstocks at ambient temperature and pressure conditions. The reaction shows high performance. The maximum Faradaic efficiency reaches 40% at the current density of 100 mA/cm². The reaction mechanism is investigated in detail. Isotope-labelled ATR-FTIR and online DEMS demonstrated the appearance of C-N, C≡N, and C=N intermediates during the electrolytic process. On the basis of experimental and DTF calculations, a possible reaction pathway going stepwise through formaldehyde, methylene imine, and hydrogen cyanide is proposed for the formation of formamide. Considering the novelty and significance of this reaction as well as the quality of the results, I recommend publishing this manuscript in *Nature Communications* after the authors address the following questions.

1) Did the authors run control experiments of aldehyde + NH₃ and acid + NH₃ to verify that the aldehyde is indeed on the reaction pathway while the acid is not?

2) Why did the authors choose 10% ¹⁵N for the isotope-labelled experiments, not fully labelled NH₃?

- 3) Did the authors perform DEMS with ^{15}N -labelled NH_3 ?
- 4) Did the authors perform ATR-FTIR for CH_3OH oxidation and NH_3 oxidation as controls?
- 5) What products account for the remaining FE?

Reviewer #3 (Remarks to the Author):

The manuscript presents an interesting approach to form C-N bond containing chemicals using CH_3OH and NH_3 . The authors presents a combined experimental and DFT study to investigate possible reaction channels for the formation of C-N bond. However, the DFT network is too simplified: no solvent effect, no reaction barrier calculations and incomplete pathways etc.

Based on the current pathways: CH_3OH undergoes de-hydrogenation reaction to form CH_2O which couples with NH_3 to form C-N bond containing intermediate. The authors need to take an comprehensive approach to discuss all competitive pathways such as but not limited to:

- CH_2O is weakly bound and may prefer to desorb

- CH_2O may undergo further dehydrogenation to form HCO which could couple with NH_x or undergoes de hydrogenation to form CO

- NH_3 may undergo dehydrogenation to form NH_2 which may couple with CH_xO intermediate to form C-N bond

Thus the authors should consider all these possibilities and present the energetics to make any firm conclusions about the reaction channel on various catalyst surfaces.

-I believe this is an electrochemical reaction, in such a case HER also needs to be considered when discussion the selectivity, faradaic efficiency.

A point-by-point response to the reviewers' comments

To reviewer 1:

Reviewer letter: In the submitted work, the group of Bin Zhang demonstrated the oxidative synthesis of formamide from NH_3 and methanol building blocks. Formamide electrosynthesis was accomplished with Pt/PtO₂ catalysts (after screening other candidates) with a high Faradaic efficiency of 40%. Mechanistic studies proposed that the key step is a nucleophilic attack by the NH_3 onto a formaldehyde like surface intermediates (originating from methanol). Finally, a techno-economic analysis was performed to point to the profitability of this route.

The area of heterogeneous electrosynthesis beyond water/ CO_2 electrolysis is rapidly growing and the discovery of new reaction pathways such as this can be a very important boost to this end. I appreciate the integration of techno-economic analysis with the theoretical and experimental efforts to drive home the importance of this. That being said, there are several issues/points to be addressed prior to consideration for publication.

Answer: We highly appreciate the reviewer for the positive and constructive comments on our communication. To save the reviewer's valuable time, key revisions are displayed in yellow background in the revised manuscript and the revised supporting information.

Comment 1. Infrared spectroscopy: The experiments here are only partially convincing. I do not understand why only 10% enriched $^{15}\text{NH}_3$ was used as 98/99% NH_3 is a readily available and cheap reagent. This should be used in place for definitive peak assignments in figure 3. Moreover, the signal quality in figure 3a especially is not sufficient for peak assignment.

If nanostructured Au was used as the substrate and I assume Pt was somehow deposited overtop, how is the activity attributed to only the Pt when Au may be exposed to the electrolyte as well. In other words, adsorbed intermediates and changes in solution that give rise to IR bands can stem from the underlying substrate as well.

Additional controls should be performed with IR measurements such as spectra taken in the absence of methanol or NH_3 .

Answer: According to the reviewer's nice suggestion, a high ^{15}N abundance of $^{15}\text{NH}_4^+$ (98.5 %), Au substrate, only methanol or only ammonia are carried out for *in situ* FTIR measurements and the

recognizable peaks have been shown in **Figure R1** (**Figure 3f in the revised manuscript and Supplementary Figure 22 in the revised SI**).

Figure R1. *In situ* electrochemical FTIR spectra under different conditions. ^{15}N isotope experiment for formamide electrosynthesis (a). Au substrate for formamide electrosynthesis (b). CH_3OH electrooxidation (c) and NH_4^+ electrooxidation (d) on Pt.

As shown in **Figure R1a**, the peaks of $\text{C}\equiv^{15}\text{N}$ at $\sim 2120\text{ cm}^{-1}$ and $\text{C}-^{15}\text{N}$ at $\sim 1360\text{ cm}^{-1}$ can be well recognized (**Figure 3f in the revised manuscript**). Additionally, $^{15}\text{NO}_3^-$ peak at $\sim 1460\text{ cm}^{-1}$ (Mol. Catal., 2018, 451, 114-124) can be also detected.

To illustrate the influence of Au substrate on the FTIR peak, Au-coated Ti foil is synthesized using the electrodeposition method to test its performance for formamide electrosynthesis at 100 mA cm^{-2} current density. As shown in **Figure R2** (**Supplementary Figure 23 in the revised SI**), no formamide is found after a three-hour reaction, suggesting its inert activity for formamide synthesis. Furthermore, as a comparison, Au-coated Si substrate is directly carried out for *in situ* FTIR (**Figure R1b, Supplementary Figure 22 in the revised SI**), water peak ($\sim 1680\text{ cm}^{-1}$), $\text{NO}_3^-/\text{CO}_3^{2-}$ ($\sim 1520\text{ cm}^{-1}$) (Mol. Catal., 2018, 451, 114-124; ACS Catal., 2019, 9, 10983-10989) and C-O (1220 cm^{-1}) bond are recognized and no C-N is found. Hence, the influence of Au substrate on C-N signal observation can be excluded.

For *in situ* FTIR measurements of single CH_3OH or NH_3 electrooxidation, CO_3^{2-} and C-O bond are detected during methanol electrooxidation and NO_3^- is detected during NH_3 electrooxidation (**Figure R1c,d, Supplementary Figure 22 in the revised SI**).

Those results prove that the FTIR peaks of Au substrate and single $\text{CH}_3\text{OH}/\text{NH}_3$ electrooxidation do not

disturb the detection of C-N and C≡N. And ^{15}N isotope experiment further demonstrates the generation of C-N and C≡N bonds.

Figure R2. ^1H -NMR spectrum for formamide electrosynthesis using Au-Ti.

Comment 2. As a reference experiment, a two chamber (e.g. H-Cell) setup should be tested as well to exclude any effects from the cathode reactivity.

Answer: We sincerely acknowledge the kind comment. A two chamber (e.g. H-Cell) separated by nafion membrane is used for studying the effect of the cathode reactivity on formamide synthesis and both sides are filled in the mixture of CH_3OH and NH_3 in 0.5M NaHCO_3 . Both cathode and anode are Pt-Ti foil. After a three-hour electrolysis at 100 mA cm^{-2} current density, formamide can only be detected at the anode side (**Figure R3, Supplementary Figure 20 in the revised SI**). The result indicates the cathode reactivity has no effect for formamide formation and formamide is generated via the electrooxidation reaction.

Figure R3. ¹H-NMR spectra for formamide electrosynthesis in the different chambers of a H-type cell separated by nafion membrane at 100 mA cm⁻².

Comment 3. Regarding the stability, the electrolyte should be tested for dissolved Pt after prolonged electrolysis and the electrode should be characterized with SEM/XRD/TEM after as well.

Answer: We sincerely acknowledge the kind comment. An inductively coupled plasma emission spectrometer (ICP) is carried out for quantifying the dissolved Pt. 1.1 ng/s dissolution of Pt is quantified at 100 mA cm⁻² current density. Although a small amount of Pt dissolution is induced by oxidation, SEM and XRD of used Pt-Ti samples still show no obvious change (**Figures R4a,b, Supplementary Figure 11 in the revised SI**). And a slight surface oxidation of used Pt is confirmed by the XPS spectrum of Pt⁴⁺ 4f_{5/2} signal peak at 78.1 eV (**Figure R4c, Supplementary Figure 11 in the revised SI**).

Figure R4. The characterization of Pt-Ti after the stability measurement. SEM image (a), XRD pattern (b), and XPS spectrum (c).

Comment 4. A schematic of the Raman experiment/reaction cell should be included in the SI.

Answer: According to the reviewer's nice suggestion, the Raman experiment/reaction cell is shown in **Figure R5** and added in the revised supporting information (**Supplementary Figure 13 in the revised SI**).

Figure R5. Schematic illustration for *in situ* Raman electrochemical measurement.

Comment 5. Regarding the measurements at low pH – formamide tends to decompose to formate and NH_3 in these conditions. Could the authors be underestimating their formamide production? The same goes for high pH, though there isn't any significant formamide production there to begin with. Is there a mechanistic explanation for the lack of formamide generation at high pH?

Answer: According to the reviewer's suggestion, we test the stability of formamide at a low pH solution (in 0.25 M H_2SO_4 , 3h) with the presence of the same amount of ammonia as the formamide electrosynthesis test. The results show no generation of formic acid. Thus, the hydrolysis of formamide can be effectively suppressed in the presence of ammonia (**Figure R6**). Hence, we calculate the yield of formamide without the consideration of formamide decomposition.

Figure R6. ^1H -NMR spectra of formamide in 0.25 M H_2SO_4 in the presence of ammonia after 3h.

At a high pH solution (0.5 M NaOH), formaldehyde as an active intermediate of methanol oxidation will be soon decomposed into formate and methanol (Cannizzaro reaction) (**Figure R7**) rather than attacked by ammonia. Thus, formamide cannot be formed a high pH solution.

Figure R7. ^1H -NMR spectrum for formaldehyde intermediate self-decomposition at high pH solution (0.5 M NaOH).

Comment 6. Does formamide form when starting from formaldehyde as a reagent? This could be important for mechanistic explanations. The same goes with formate.

Answer: According to the reviewer's nice suggestion, those two control experiments are carried out. As shown from **Table R1 (Supplementary Table S6 in the revised SI)**, if the N source (NH_3) keeps constant and CH_3OH feedstock is replaced with HCOH or HCOO^- , formamide can be only detected in the mixture of HCOH and NH_3 , proving the importance of *in situ* formed aldehyde-like intermediate.

Table R1. Control experiments for exploring the reaction pathway.

Entry	C Source	N Source	Current density / mA cm^{-2}	Electricity	Formamide
1	CH_3OH	NH_3	100	NaHCO_3	Yes
2	HCOH	NH_3	100	NaHCO_3	Yes
3	HCOO^-	NH_3	100	NaHCO_3	No

Comment 7. The authors cited a work detailing acetamide formation from CO and NH_3 electrolysis. I would also recommend referencing works that discussed formamide production such as: *Chemical Science*, 13, 3957-3964 (2022).

In general, a discussion regarding previous C-N bond formation in electrochemistry would be useful to add in the introduction as well as some text regarding how this work conceptually advances the state of the art of our understanding of this process.

Answer: According to the reviewer's nice suggestion, a reference work about C-N electroreduction synthesis especially for formamide is described and cited. "*the electrosynthesis of methylamine, formamide, and acetamide has been successfully achieved by using CO_2 and CO as the carbon source (Chem. Sci. 2022, 13, 3957-3964.; Nat. Rev. Chem. 2022, 6, 303-319.; Nat. Sustain. 4, 2021, 725-730.; Nat. Chem. 2019, 11, 846-851.), respectively. At present, these important advances mainly focused on the*

electrochemical reduction reactions to construct the C-N bond. But, the sluggish anodic oxygen evolution reaction requires a high applied potential. Thus, the exploration of an alternative electrooxidation process using abundant carbon- and nitrogen-containing feedstocks to synthesize formamide under ambient conditions is attractive but remains a great challenge”.

Comment 8. Regarding the scope of this scheme, if another amine was used in place of ammonia, could the same nucleophilic attack and C-N bond formation occur?

Answer: According to the reviewer’s suggestion, methylamine is used in place of ammonia and formyl methylamine can be also synthesized. The FEs for formyl methylamine under the different current densities are shown in **Figure R8 (Supplementary Figure 12 in the revised SI)**. The highest Faradaic efficiencies can reach 20.54 %.

Figure R8. Current density-dependent Faradaic efficiency and yield rate of HCONCH₃ over Pt-Ti catalyst.

Comment 9. For the mechanism proposed by the authors, is cyanide ever detected in the solution as it is an intermediate. What about methylamine? The full NMR spectra should be shown, not only the formate/formamide region.

Answer: As shown from the typical ¹H-NMR spectrum, no cyanide and methylamine are detected (**Figure R9**), because they will be fast consumed in the next step.

Figure R9. The typical ^1H -NMR spectrum for formamide electrosynthesis.

To reviewer 2:

Reviewer letter: In this work, Zhang et al. developed a sustainable electrooxidation approach to synthesize formamide using methanol and ammonia as feedstocks at ambient temperature and pressure conditions. The reaction shows high performance. The maximum Faradaic efficiency reaches 40% at the current density of 100 mA/cm². The reaction mechanism is investigated in detail. Isotope-labelled ATR-FTIR and online DEMS demonstrated the appearance of C-N, C≡N, and C=N intermediates during the electrolytic process. On the basis of experimental and DFT calculations, a possible reaction pathway going stepwise through formaldehyde, methylene imine, and hydrogen cyanide is proposed for the formation of formamide. Considering the novelty and significance of this reaction as well as the quality of the results, I recommend publishing this manuscript in Nature Communications after the authors address the following questions.

Answer: We thank the reviewer for the positive and constructive comments on our communication. To save the reviewer's valuable time, key revisions are displayed in a yellow background in the revised manuscript and Supporting Information.

Comment 1. Did the authors run control experiments of aldehyde + NH₃ and acid + NH₃ to verify that the aldehyde is indeed on the reaction pathway while the acid is not?

Answer: According to the reviewer's nice suggestion, those two control experiments are carried out. As shown in **Table R1 (Supplementary Table S6 in the revised SI)**, if the N source (NH₃) keeps constant and CH₃OH feedstock is replaced with HCOH or HCOO⁻, formamide can be only detected in the mixture of HCOH and NH₃, proving the importance of *in situ* formed aldehyde-like intermediate.

Table R1. Control experiments for exploring the reaction pathway.

Entry	C Source	N Source	Current density / mA cm ⁻²	Electricity	Formamide
1	CH ₃ OH	NH ₃	100	NaHCO ₃	Yes

2	HCOH	NH ₃	100	NaHCO ₃	Yes
3	HCOO ⁻	NH ₃	100	NaHCO ₃	No

Comment 2. Why did the authors choose 10% ¹⁵N for the isotope-labelled experiments, not fully labelled NH₃?

Answer: According to the reviewer's nice suggestion, a high ¹⁵N abundance of ¹⁵NH₄⁺ (98.5 %), Au substrate, only methanol or only ammonia are carried out for *in situ* FTIR measurements and the recognizable peaks have been shown in **Figure R10** (**Figure 3f in the revised manuscript and Supplementary Figure 22 in the revised SI**).

Figure R10. *In situ* electrochemical FTIR spectra under the different conditions. ¹⁵N isotope experiment for formamide electro synthesis (a). Au substrate for formamide electro synthesis (b). CH₃OH electrooxidation (c) and NH₄⁺ electrooxidation (d) on Pt.

As shown in **Figure R10a**, the peaks of C≡¹⁵N at ~2120 cm⁻¹ and C-¹⁵N at ~1360 cm⁻¹ can be well recognized (**Figure 3f in the revised manuscript**). Additionally, ¹⁵NO₃⁻ peak at ~1460 cm⁻¹ (Mol. Catal., 2018, 451, 114-124) is also detected.

To illustrate the influence of Au substrate on the FTIR peak, Au-coated Ti foil is synthesized using the electrodeposition method to test its performance for formamide electro synthesis at 100 mA cm⁻² current

density. As shown in **Figure R11 (Supplementary Figure 23 in the revised SI)**, no formamide is found after a three-hour reaction, suggesting its inert activity for formamide synthesis. Furthermore, as a comparison, Au-coated Si substrate is directly carried out for *in situ* FTIR (**Figure R10b, Supplementary Figure 22 in the revised SI**), water peak ($\sim 1680\text{ cm}^{-1}$), $\text{NO}_3^-/\text{CO}_3^{2-}$ ($\sim 1520\text{ cm}^{-1}$) (Mol. Catal., 2018, 451, 114-124; ACS Catal., 2019, 9, 10983-10989) and C-O (1220 cm^{-1}) bond are recognized and no C-N is found. Hence, the influence of Au substrate on C-N signal observation can be excluded.

For *in situ* FTIR measurements of single CH_3OH or NH_3 electrooxidation, CO_3^{2-} and C-O bond are detected during methanol electrooxidation and NO_3^- is detected during NH_3 electrooxidation (**Figure R10c,d, Supplementary Figure 22 in the revised SI**).

Those results prove that the FTIR peaks of Au substrate and single $\text{CH}_3\text{OH}/\text{NH}_3$ electrooxidation do not disturb the detection of C-N and $\text{C}\equiv\text{N}$. And ^{15}N isotope experiment further demonstrates the generation of C-N and $\text{C}\equiv\text{N}$ bonds.

Figure R11. ^1H -NMR spectrum for formamide electrosynthesis using Au-Ti.

Comment 3. Did the authors perform DEMS with ^{15}N -labelled NH_3 ?

Answer: We sincerely acknowledge the kind comment. If DEMS is carried out using ^{15}N -labelled NH_3 , the mass signal of the key intermediate ($\text{H}_2\text{C}=\text{}^{15}\text{NH}$) will overlap with the signal of HCOH and $^{15}\text{N}_2$. Hence, DEMS with ^{15}N -labelled NH_3 is not performed.

Comment 4. Did the authors perform ATR-FTIR for CH_3OH oxidation and NH_3 oxidation as controls?

Answer: According to the reviewer's nice suggestion, *in situ* FTIR measurements are carried out in the absence of methanol or NH_3 . For *in situ* FTIR measurements of single CH_3OH or NH_3 electrooxidation (**Figure R10c,d**), CO_3^{2-} and C-O bond are detected during methanol electrooxidation and NO_3^- is detected during NH_3 electrooxidation. Those results prove that the FTIR peaks of single CH_3OH or NH_3 electrooxidation do not disturb the detection of C-N and $\text{C}\equiv\text{N}$.

Comment 5. What products account for the remaining FE?

Answer: Besides formamide and formate, O_2 , N_2 , CO_2 , NO_3^- and NO_2^- are produced (**Figure R12**, **Supplementary Figure 8 in the revised SI**).

Figure R12. The typical distribution of the products over formamide electro-synthesis.

To reviewer 3:

Reviewer letter: The manuscript presents an interesting approach to form C-N bond containing chemicals using CH₃OH and NH₃. The authors presents a combined experimental and DFT study to investigate possible reaction channels for the formation of C-N bond. However, the DFT network is too simplified: no solvent effect, no reaction barrier calculations and incomplete pathways etc.

Answer: We thank the reviewer for the positive and constructive comments on our communication. To save the reviewer's valuable time, key revisions are displayed in a yellow background in the revised manuscript and Supporting Information.

Comment 1. "no solvent effect "

Answer: We sincerely acknowledge the kind comment. The solvation effect of water is included in all the simulations of surfaces and adsorbates through the implicit solvation model and VASPSol code developed by Hennig et al. (*J. Chem. Phys.* 2014, 140, 084106; *J. Chem. Phys.* 2019, 151, 234101). The results of the water solvation effect are summarized in the following Table and added in **Table R2**.

Table R2. The water solvation energy of various adsorbates on the surfaces of α -PtO₂, β -NiOOH, and α -FeOOH.

α -PtO ₂							
Adsorbates	*CH ₃ OH	*CH ₂ OH	*CH ₂ O	*HCO	*NH ₃	*NH ₂	*NH
E_{sol}/eV	-0.20	-0.26	-0.13	-0.10	-0.36	-0.12	-0.14
Adsorbates	*CH ₂ OHNH ₂	*CH ₂ ONH ₂	*CHONH ₂	*CH ₂ NH	*CHNH	*CHN	
E_{sol}/eV	-0.28	-0.15	-0.20	-0.24	-0.13	-0.17	
β -NiOOH							
Adsorbates	*CH ₃ OH	*CH ₂ OH	*CH ₂ O	*HCO	*NH ₃	*NH ₂	*NH
E_{sol}/eV	-0.08	0.11	0.08	-0.10	0.05	0.07	0.07

Adsorbates	*CH ₂ OHNH ₂	*CH ₂ ONH ₂	*CHONH ₂	*CH ₂ NH			
E_{sol}/eV	-0.24	-0.17	-0.13	0.03			
α -FeOOH							
Adsorbates	*CH ₃ OH	*CH ₂ OH	*CH ₂ O	*HCO	*NH ₃	*NH ₂	*NH
E_{sol}/eV	0.14	0.24	0.30	-0.10	0.10	0.30	0.19
Adsorbates	*CH ₂ OHNH ₂	*CH ₂ ONH ₂	*CHONH ₂	*CH ₂ NH			
E_{sol}/eV	-0.01	0.26	0.32	0.23			

The following sentence was added to the theoretical method parts in the revised Supporting information: “The solvation effect of water is included in all the simulations of surfaces and adsorbates through the implicit solvation model and VASPSol code developed by Hennig et al. (*J. Chem. Phys.* 140, 084106 (2014); *J. Chem. Phys.* 151, 234101 (2019)) and the solvation energy is taken into account of Gibbs free energy of adsorbates and reactions”.

Comment 2. ‘no reaction barrier calculations’

Answer: We sincerely acknowledge the kind comment. For the C-N bond-making steps, the thermodynamic feasibility between NH_x and *CH_xO on the surfaces of α -PtO₂, β -NiOOH, and α -FeOOH surfaces is summarized in **Table R3** as the following. Since methanol and ammonia coupling reaction takes place under highly oxidized potentials (> ~1.5 V vs. RHE), the hydrogenation through the transfer of (H⁺+e) would be hardly possible. We consider once the formation of *NH and *CO, there is no chance for such species to participate in the C-N bond-making steps but only in the byproduct formation. There are three pathways for amide formation using alcohol and ammonia as the feedstocks on α -PtO₂: (Path 1) aldehyde from alcohol dehydration reacted with NH₃ to form hemiaminal that is subsequently dehydrated to formamide; (Path 2) aldehyde as the intermediate of methanol oxidation reacts with NH₃ to form aldimine via a hemiaminal intermediate, and then the aldimine is oxidized to nitrile that can be further hydrolyzed to formamide; (Path 3) alcohol-derived CHO* reacts with NH₃-derived NH₂* to generate formamide (**Supplementary Fig. 14, Supplementary Table 1 in the revised SI**). For Path 3, the

C-N coupling requires two adjacent active intermediates of CHO^* and NH_2^* , which are both easy to be solely oxidized into the corresponding carbon-/nitrogen-containing byproducts. Thus, the direct nucleophilic attack of NH_3 on the *in situ* formed aldehyde in Paths 1 and 2 seems more possible (**Fig. 3d in the revised manuscript**). As for the C-N bond making between $^*\text{CH}_2\text{O}$ and $^*\text{NH}_2$, we suppose such a process hardly occurs due to the steric blocking of the surface.

Table R3. The reaction energy of the C-N bond-making steps on $\alpha\text{-PtO}_2$, $\beta\text{-NiOOH}$, and $\alpha\text{-FeOOH}$ surfaces.

	Reaction energy of C-N bond-making steps / eV			
	$^*\text{CH}_2\text{O-NH}_3$	$^*\text{CH}_2\text{O-}^*\text{NH}_2$	$^*\text{HCO-NH}_3$	$^*\text{HCO-}^*\text{NH}_2$
$\alpha\text{-PtO}_2$	-1.12	-0.39	0.32	-2.13
$\beta\text{-NiOOH}$	-1.38	0.36	-	-
$\alpha\text{-FeOOH}$	-2.25	1.72	0.11	0.24

*HCO on $\beta\text{-NiOOH}$ surface automatically forms HCOO group.

For the possible C-N bond-making steps, such as $^*\text{CH}_2\text{O}+\text{NH}_3=^*\text{CH}_2\text{OHNH}_2$, we have tried to implement the typical Nudged Elastic Band (NEB) and/or ci-NEB method for searching transition states and calculating barrier energies. **Figure R13 (Supplementary Figure 19 in the revised SI)** shows the movie-images of the barrier path with $^*\text{CH}_2\text{O}+\text{NH}_3$ on $\alpha\text{-PtO}_2$ surface as an example.

Figure R13. The initial state (IS), final state (FS), and movie images for searching transition states using NEB method.

However, these calculations hardly converge. The weak interaction between $^*\text{CH}_2\text{O}$ / NH_3 and the surfaces means that the image-species, especially the H atom that transfers from NH_3 to CH_2O have very weak sensitivity to the surface electronic structures as well as have multi-dimensional freedom to move, which twists the whole strings during the calculations. The other reason for the failure in convergence could be the difference between IS and FS are relatively large (minimum 6 images to be inserted to give a reasonable path). Therefore, we give up the NEB methods to calculate barrier energies. Instead, the C-N bond-making step between $^*\text{CH}_2\text{O}$ and NH_3 is the nucleophilic attack process, i.e., the positively charged C is attacked by the electronegative N atom in NH_3 (*Angew. Chem. Int. Ed.* 51, 544-547 (2012)). Therefore, the charged states of C in $^*\text{CH}_x\text{O}$ and N in NH_3/NH_2 qualitatively demonstrate the feasibility of the C-N bond-making process. The charge analysis of the relevant adsorbates is done using Bader charge analysis. The charged states of the C atom and N atom are presented below (**Table R4, Supplementary Table S4 in the revised SI**):

Table R4. The charged states of C atom and N atom in the relevant adsorbates on $\alpha\text{-PtO}_2$, $\beta\text{-NiOOH}$, and $\alpha\text{-FeOOH}$ surfaces.

	C in $^*\text{CH}_2\text{O}$	N in NH_3
$\alpha\text{-PtO}_2$	+1.60	-3.00
$\beta\text{-NiOOH}$	+1.36	-3.00
$\alpha\text{-FeOOH}$	+0.20	-3.00

As shown in **Table R4**, C in $^*\text{CH}_2\text{O}$ or $^*\text{CHO}$ is positively charged in the order of $+1.60\text{ e}$ on $\alpha\text{-PtO}_2 > +1.36\text{ e}$ on $\beta\text{-NiOOH} > +0.20\text{ e}$ on $\alpha\text{-FeOOH}$. We, therefore, propose that the barrier energies for the C-N bond-making process by NH_3 nucleophilic attack of $^*\text{CH}_2\text{O}$ are very likely to be in the order of $\alpha\text{-PtO}_2 > \beta\text{-NiOOH} > \alpha\text{-FeOOH}$.

Comment 3. 'incomplete pathways'

Answer: We sincerely acknowledge the kind comment. The reaction network including various electrochemical dehydrogenation steps as well as C-N bond-making steps is presented under a uniform applied potential of 1.68 V vs RHE, at which the generation of $^*\text{NH}_2$ on $\alpha\text{-PtO}_2$ becomes thermodynamically feasible. It could be noticed through **Figures R14-R16 (Supplementary Figures 14-16)**

that the production of formamide has the highest thermodynamic driving force on α -PtO₂ and the lowest driving force for byproduct formation, which are summarized in **Table R5 (Supplementary Table S5 in the revised SI)** and explained in the main text 'For Path 1, the minimum applied potentials demanded to make all elementary reactions exoergic are 1.00 V, 2.51 V, and 2.57 V for α -PtO₂, β -NiOOH, and α -FeOOH, respectively (**Table R5**). α -PtO₂ performs an obviously stronger catalytic activity toward the coupling of methanol and ammonia, which is rooted in its higher ability of oxidizing methanol and stabilizing *CH₂ONH₂ (**Table R5**). It should be noted that α -PtO₂ also benefits the suppression of the complete methanol oxidation due to its smallest driving force of the *CH₂O-*CHO step (**Table R5**). For Path 2, the crucial reaction step of *CH₂OHNH₂ dehydration only thermodynamically takes place on α -PtO₂ with Gibbs free energy of -0.19 eV, which excludes the possibility of Path 2 on β -NiOOH and α -FeOOH (**Table R5**)'.

Figure R14. The reaction network of methanol and ammonia under an applied potential of 1.68 V vs RHE on α -PtO₂. The formation of *NH and *CO leads to the byproduct generation of N₂, nitrate, nitrate, CO₂, and acetate. All the reaction steps connected by dotted lines are the electrochemical dehydration processes with the transfer of (H⁺ + e).

Figure R15. The reaction network of methanol and ammonia under an applied potential of 1.68 V vs RHE on β -NiOOH. The formation of *NH and *CO leads to the byproduct generation of N_2 , nitrate, nitrate, CO_2 , and acetate. All the reaction steps connected by dotted lines are the electrochemical dehydration processes with the transfer of ($H^+ + e$).

Figure R16. The reaction network of methanol and ammonia under an applied potential of 1.68 V vs RHE on α -FeOOH. The formation of *NH and *CO leads to the byproduct generation of N_2 , nitrate, nitrate, CO_2 , and acetate. All the reaction steps connected by dotted lines are the electrochemical dehydration processes with the transfer of ($H^+ + e$).

Table R5. The summarized properties of α -PtO₂, β -NiOOH, and α -FeOOH as the catalyst for the coupling of methanol and ammonia.

	Applied potential for CH ₃ OH-*CH ₂ O step / V	Adsorption energy of *CH ₂ ONH ₂ / eV	Reaction energy of *CH ₂ O-*CHO at 0 V / eV	Reaction energy of *CH ₂ OHNH ₂ dehydration / eV	Potential dependent step & Reaction energy / eV
α -PtO ₂	0.80	-0.23	-0.13	-0.19	*CH ₂ OH → *CH ₂ O & 1.00
β -NiOOH	1.00	0.91	-2.03	0.55	*H ₂ NCH ₂ OH → *H ₂ NCH ₂ O & 2.51
α -FeOOH	2.25	1.80	-0.73	1.08	*H ₂ NCH ₂ O → *H ₂ NCHO & 2.57

The reaction pathway and the energy diagram for formamide formation on β -NiOOH and α -FeOOH at applied potentials of 2.51 V vs. RHE and 2.57 V vs. RHE, respectively, are shown in **Figure R17 (Supplementary Figure S17 in the revised SI)** and **Figure R18 (Supplementary Figure S18 in the revised SI)**.

Figure R17. The reaction pathway and the energy diagram for formamide formation on β -NiOOH at applied potentials of 2.51 V vs RHE. All steps in the shown pathway are thermodynamically feasible.

Figure R18. The reaction pathway and the energy diagram for formamide formation on α -FeOOH at applied potentials of 2.57 V vs RHE. All steps in the shown pathway are thermodynamically feasible.

Comment 4. Based on the current pathways: CH_3OH undergoes de-hydrogenation reaction to form CH_2O which couples with NH_3 to form C-N bond containing intermediate. The authors need to take a comprehensive approach to discuss all competitive pathways such as but not limited to: CH_2O is weakly bound and may prefer to desorb; CH_2O may undergo further dehydrogenation to form HCO which could couple with NH_x or undergoes dehydrogenation to form CO ; NH_3 may undergo dehydrogenation to form NH_2 which may couple with CH_xO intermediate to form C-N bond. Thus the authors should consider all these possibilities and present the energetics to make any firm conclusions about the reaction channel on various catalyst surfaces.

Answer: We thank the reviewer's comment.

Firstly, $^*\text{CH}_2\text{O}$ is bonded to the surface with the adsorption energy of -0.20 eV at 0 V with the reference of CH_2O molecular in a vacuum. Therefore, the desorption of $^*\text{CH}_2\text{O}$ is scarce, which is confirmed by the experimental results that no formaldehyde is detected in the solution.

Secondly, $^*\text{CH}_2\text{O}$ will indeed undergo dehydrogenation to form $^*\text{HCO}$ on all three types of surfaces. Only on α - PtO_2 surface, C-N bond making between $^*\text{HCO}$ and NH_3 occurs. On α - FeOOH and β - NiOOH , $^*\text{HCO}$ only leads to the formation of byproducts. The detailed reaction pathway and the energy diagram are shown in **Figures R14-R18** and **Figure 3 in the revised manuscript** and the thermodynamics analysis of the C-N bond-making steps is summarized in **Table R3**.

Thirdly, NH_3 will also undergo dehydrogenation to form $^*\text{NH}_2$ on all three types of surfaces. The C-N bond making between $^*\text{HC}_x\text{O}$ and $^*\text{NH}_2$ only thermodynamically occurs on α - PtO_2 . On α - FeOOH and β - NiOOH , $^*\text{HCO}$ leads to the formation of byproducts. The detailed reaction pathway and the energy diagram are shown in **Figures R14-R18** and **Figure 3 in the revised manuscript** and the thermodynamics analysis of the C-N bond-making steps is summarized in **Table R2**.

The possibilities of the formation of other adsorbates, such as $^*\text{CO}$, $^*\text{NH}$, etc., are presented in **Figures R14-R16**.

Comment 5. I believe this is an electrochemical reaction, in such a case HER also needs to be considered when discussing the selectivity, faradaic efficiency.

Answer: HER does not occur in the anode electrode. In our manuscript, we reported an electrooxidation strategy to synthesize formamide from methanol and ammonia. We think what the reviewer considers is OER. OER has been considered when we discuss selectivity and faradaic efficiency.

Department of Chemistry
Tianjin University
Tianjin 300072, P. R. China
E-mail: bzhang@tju.edu.cn

We acknowledge all the kind comments and wise suggestions from three reviewers. We are sure that the quality of this work will be greatly improved according to these helpful comments and wise suggestions.

We are sure that the quality of this work will be greatly improved according to these nice comments and wise suggestions.

REVIEWER COMMENTS

Reviewer #1 (Remarks to the Author):

The authors have addressed most points. Only some minor revisions and a reconsideration/re-evaluation of band assignments should be performed.

1) I would double check the assignments for the nitrate band. NO_3^- has its main absorption peak around 1350 cm^{-1} . The authors should take spectra of several standards of NO_3 , NO_2 to compare. The band assigned to the C-N vibration is likely nitrate. The reference cited in the figure caption primarily looks at NO decomposition rather than being an absolute reference for NO_3 .

2) In figure R1-d, the band at 15550 cm^{-1} is likely not nitrate (again, the authors can verify with a standard of nitrate to see) and the signatures assigned to the bands are just from the vibrational/rotational signature of water. In figure R1 – b and c the symmetric signal below and above 1640 is also likely just from water. The authors should verify by testing the system without any methanol or ammonia.

Point 1) and 2) should be incorporated into figure 3 of the main text in figure 3 e and f. The band assignments here are likely to be wrong. Again, NO_3^- – at 1350 cm^{-1} would also have a shift if 15NH_4 is used. The water band is likely the positive peak at 1640 cm^{-1} and the symmetric signal around it is likely from its rotational modes. This can also be verified with D_2O used instead...

4) For table R1 (S6) – the production formamide should be quantified (faradaic efficiency, partial current density...)

Reviewer #3 (Remarks to the Author):

The authors have addressed most of my comments.

Lines 148-149: a correlation between charge and activation barrier has been made. What is the origin of such correlation?

It is difficult to follow the manuscript as one has to go back and forth to SI. The authors need to present a figure summarizing all possible reaction pathways in the main text. Also the energetics of reaction pathways on different catalyst surfaces need to be included in the main text. Authors can think of presenting free energy diagrams in the main text: one free energy diagram along three reaction channels on one catalyst surface and so on....

A point-by-point response to the reviewers' comments

To reviewer 1:

Reviewer letter: The authors have addressed most points. Only some minor revisions and a reconsideration/re-evaluation of band assignments should be performed.

Answer: We highly appreciate the reviewer for the positive comments on our communication. To save the reviewer's valuable time, key revisions are displayed in yellow background in the revised manuscript and the revised supporting information.

Comment 1. I would double check the assignments for the nitrate band. NO_3^- has its main absorption peak around 1350 cm^{-1} . The authors should take spectra of several standards of NO_3^- , NO_2^- ... to compare. The band assigned to the C-N vibration is likely nitrate. The reference cited in the figure caption primarily looks at NO decomposition rather than being an absolute reference for NO_3^- .

Answer: We sincerely acknowledge the kind comment. According to the reviewer's suggestion, we first measure the FTIR spectrum of NO_3^- . As the reviewer claimed, NO_3^- shows an absorption at around $1350\text{-}1380\text{ cm}^{-1}$ (**Figures R1a,b**). But, we cannot find a peak at around $1350\text{-}1380\text{ cm}^{-1}$ in the cutting-edge ATR-FTIR of $^{14}\text{NH}_4^+$ and methanol electro-oxidation (**Figure 3e in the revised manuscript**), indicating that the *in-situ* generated small amounts of NO_3^- is not sensitive in our measurement. Thus, 1350 cm^{-1} in the ATR-FTIR of $^{15}\text{NH}_4^+$ and methanol electro-oxidation (**Figure 3f in the revised manuscript**), should not be ascribed to $^{15}\text{NO}_3^-$. Moreover, we thank the reviewer for pointing the issue out, the peak at 1520 cm^{-1} in **Figure 1e in the revised manuscript** and **Supplementary Figure 22 in the revised SI** should be ascribed to the adsorbed NO intermediate (a common intermediate for NH_3 -to- NO_3^- conversion, Green Chem. 2022, 24, 1578-1589).

Figure R1. FTIR spectra of (a) H₂O, (b) H₂O + NO₃⁻. (c) *In situ* electrochemical FTIR spectra under different current densities in the electrolyte without methanol and ammonia.

Comment 2. In figure R1-d, the band at 1550 cm⁻¹ is likely not nitrate (again, the authors can verify with a standard of nitrate to see) and the signatures assigned to the bands are just from the vibrational/rotational signature of water. In figure R1 - b and c the symmetric signal below and above 1640 is also likely just from water. The authors should verify by testing the system without any methanol or ammonia.

Answer: As mentioned above, the peak at 1520 cm⁻¹ in **Figure 1e in the revised manuscript** and **Supplementary Figure 22c in the revised SI** should be the adsorbed NO intermediate (Green Chem. 2022, 24, 1578-1589). According to the reviewer's suggestion, we have tested the *in situ* FTIR spectra without methanol and ammonia (**Figure R1c**). Only one peak appears at around 1660 cm⁻¹, which is consistent with previous reports (J. Am. Chem. Soc. 2022, 144, 2079-2084; ACS Energy Lett. 2019, 4, 682-689; Nat. Commun. 2021, 12, 5745). It implies that the symmetric signal below and above 1640 cm⁻¹ in **Figures 22a,b in the revised SI** is not from water.

Comment 3. In figure R1-d, the band at 1550 cm⁻¹ is likely not nitrate (again, the authors can verify with

a standard of nitrate to see) and the signatures assigned to the bands are just from the vibrational/rotational signature of water. In figure R1 b and c the symmetric signal below and above 1640 is also likely just from water. The authors should verify by testing the system without any methanol or ammonia. Point 1) and 2) should be incorporated into figure 3 of the main text in figure 3 e and f. The band assignments here are likely to be wrong. Again, NO_3^- at 1350 cm^{-1} would also have a shift if $^{15}\text{NH}_4$ is used. The water band is likely the positive peak at 1640 cm^{-1} and the symmetric signal around it is likely from its rotational modes. This can also be verified with D_2O used instead.

Answer: As mentioned above, the peak at around 1520 cm^{-1} in **Figure 1e in the revised manuscript** is the adsorbed NO intermediate. According to the reviewer's suggestion, we have tested the FTIR spectra without methanol and ammonia. As seen from **Figure R1c**, only one peak appears at around 1660 cm^{-1} , which is consistent with the previous report (J. Am. Chem. Soc. 2022, 144, 2079-2084; ACS Energy Lett. 2019, 4, 682-689; Nat. Commun. 2021, 12, 5745). Thus, the symmetric signal below and above 1640 cm^{-1} is not from water. According to the review's suggestion, we have revised the **Figures 3e,f in the revised manuscript** as follows:

"Using $^{14}\text{NH}_4^+$ as the nitrogen source, the signal shows an increasing tendency with the current density (Fig. 3e). The characteristic peaks at around 1220 cm^{-1} , 1435 cm^{-1} , 1520 cm^{-1} , 1680 cm^{-1} , and 2185 cm^{-1} , corresponding to C-O, C-N, $\text{CO}_3^{2-}/\text{NO}$, H_2O , and $\text{C}\equiv\text{N}$ can be identified, suggesting the production of multiple intermediates. For confirming the C-N bond formation, isotope-labelling in situ ATR-FTIR is carried out using $^{15}\text{NH}_4^+$ (Fig. 3f). By applying the increasing current density, $\text{C}-^{14}\text{N}$ and $\text{C}\equiv^{14}\text{N}$ shift to the lower wavenumber region (1360 cm^{-1} for $\text{C}-^{15}\text{N}$, and 2120 cm^{-1} for $\text{C}\equiv^{15}\text{N}$) because of the isotope effect. ^{15}NO signal at around 1460 cm^{-1} is also observed."

Comment 4. For table R1 (S6) - the production formamide should be quantified (faradaic efficiency, partial current density...)

Answer: According to the reviewer's kind suggestion, the production formamide (faradaic efficiency, partial current density) is quantified and shown in **Table R1 (Table S6 in the revised SI)**.

Table R1. Control experiments for exploring the reaction pathway.

Entry	C Source	N Source	Current density / mA cm ⁻²	Electricity	Formamide / (Faradaic efficiency / Partial current density)
1	CH ₃ OH	NH ₃	100	NaHCO ₃	32.70 % / 32.70 mA cm ⁻²
2	HCOH	NH ₃	100	NaHCO ₃	4.11 % / 4.11 mA cm ⁻²
3	HCOO ⁻	NH ₃	100	NaHCO ₃	Null

To reviewer 2:

Reviewer letter: The authors have addressed most of my comments.

Answer: We highly appreciate the reviewer for the positive and constructive comments on our communication. To save the reviewer's valuable time, key revisions are displayed in yellow background in the revised manuscript and the revised supporting information.

Comment 1. Lines 148-149: a correlation between charge and activation barrier has been made. What is the origin of such correlation?

Answer: We sincerely acknowledge the kind comment. The C-N bond formation step is through the nucleophilic attack of NH_3 on the *in situ* formed formaldehyde-like ($^*\text{CH}_2\text{O}$) intermediate on various surfaces. It has been widely recognized that the electrophilicity of carbon in the carbonyl substrate increases the reaction rate [A. Williams Concerted Organic and Bio-Organic Mechanism CRC, Boca Raton (2000) Chapter 4; A.J. Kirby C.H. Bamford, C.F.H. Tipper (Eds.), Comprehensive Chemical Kinetics, Elsevier, Amsterdam (1980), p. 161; K.A. Connors Structure Reactivity Relationships: the Study of Reaction Rates in Solution VCH, New York (1990) p 311; Michael B. Smith March's Advanced Organic Chemistry: Reactions, Mechanisms, and Structure, 8th Edition (2019), chapter 10.7.1.; <https://doi.org/10.1016/j.tet.2004.10.085>]. Considering that the reaction mechanism as well as the reaction environment of the nucleophilic attack process on the surfaces of PtO_2 , NiOOH , and FeOOH are the same, we propose that the electrophilicity or the charged state of carbon has a positive correlation with the activation barrier of the nucleophilic attack process, i.e., the higher the positive charge on carbon center, the lower the activation barrier.

Comment 2. It is difficult to follow the manuscript as one has to go back and forth to SI. The authors need to present a figure summarizing all possible reaction pathways in the main text. Also the energetics of reaction pathways on different catalyst surfaces need to be included in the main text. Authors can think of presenting free energy diagrams in the main text: one free energy diagram along three reaction channel on one catalyst surface and so on

Answer: According to the reviewer's kind suggestion, free energy diagram of HCONH_2 formation over $\alpha\text{-PtO}_2$, $\beta\text{-NiOOH}$ and $\alpha\text{-FeOOH}$ is added in **Figure R2 (Figure 3d in the revised manuscript)**. The corresponding description was revised as follows:

“The complete reaction pathway and energy diagram of the coupling reaction as well as the electronic analysis of the C-N bond formation steps are displayed in Fig. 3d”

Figure R2. The superficial active sites, mechanism, and reaction intermediates analysis. (a-c) The current density-dependent Raman signals of Pt, Ni, and Fe. (d) The theoretical model of α -PtO₂, formamide formation pathway over its surface and free energy diagram of HCONH₂ formation over α -PtO₂, β -NiOOH and α -FeOOH via different pathways. (e,f) Isotope-labeled *in situ* ATR-FTIR

Department of Chemistry
Tianjin University
Tianjin 300072, P. R. China
E-mail: bzhang@tju.edu.cn

measurements using $^{14}\text{NH}_4^+$ and $^{15}\text{NH}_4^+$.

We acknowledge all the kind comments and wise suggestions from three reviewers. We are sure that the quality of this work will be greatly improved according to these helpful comments and wise suggestions.